# Novel feature extraction method for signal analysis based on independent component analysis and wavelet transform

**Mariusz Topolski** *, **Jędrzej Kozal**

Department of Systems and Computer Networks, Faculty of Information and Communication Technology, Wrocław University of Science and Technology, Wrocław, Poland

☯ These authors contributed equally to this work.
\* mariusz.topolski@pwr.edu.pl

## Abstract

Feature extraction is an important part of data processing that provides a basis for more complicated tasks such as classification or clustering. Recently many approaches for signal feature extraction were created. However, plenty of proposed methods are based on convolutional neural networks. This class of models requires a high amount of computational power to train and deploy and large dataset. Our work introduces a novel feature extraction method that uses wavelet transform to provide additional information in the Independent Component Analysis mixing matrix. The goal of our work is to combine good performance with a low inference cost. We used the task of Electrocardiography (ECG) heartbeat classification to evaluate the usefulness of the proposed approach. Experiments were carried out with an MIT-BIH database with four target classes (Normal, Vestibular ectopic beats, Ventricular ectopic beats, and Fusion strikes). Several base wavelet functions with different classifiers were used in experiments. Best was selected with 5-fold cross-validation and Wilcoxon test with significance level 0.05. With the proposed method for feature extraction and multi-layer perceptron classifier, we obtained 95.81% BAC-*score*. Compared to other literature methods, our approach was better than most feature extraction methods except for convolutional neural networks. Further analysis indicates that our method performance is close to convolutional neural networks for classes with a limited number of learning examples. We also analyze the number of required operations at test time and argue that our method enables easy deployment in environments with limited computing power.

## 1 Introduction

Signal processing is a rapidly developing field. With an abundance of new data and the development of user electronics, demand for methods that can quickly analyze incoming data increases. Both cloud and edge computing approaches can address these problems. The speed and accuracy of algorithms are crucial in cloud and edge solutions. For this reason, we introduce a novel method for signal feature extraction that obtains performance slightly below best-

**Data Availability Statement:** All relevant data are within the paper.

**Funding:** This work is supported by the CEUS-UNISONO programme, which has received funding

from the National Science Centre, Poland under grant agreement No. 2020/02/Y/ST6/00037.

**Competing interests:** NO authors have competing interests.

performing methods and requires less compute at test time. To show this, we compare the number of operations required by our method with the number of operations in commonly used methods for signal classification. To evaluate the performance of our method an ECG task was selected, as it is an important problem with well-established datasets and evaluation procedures [1, 2]. Also, there are already well-performing algorithms developed specifically for this field [3–5].

## 1.1 Signal feature extraction

Independent component analysis (ICA) [6] is a statistical method that solves the problem of blind source separation. In [7] ICA was connected with a neural network for ECG arrhythmia classification. Authors constructed feature vectors from ICA and RR intervals. RR interval is an ECG signal fragment cropped from one R wave peak to the next one. As classifiers, probabilistic neural networks and backpropagation neural networks were utilized. In [8] authors used wavelet transforms and ICA for feature extraction as separate components of feature vectors. Similarly to [7] RR intervals were also included as a part of a feature vector. Another method for signal feature extraction is a *discrete wavelet transform (DWT)*. It can be utilized for the processing of various signals. In [9] DWT was used for electromyography (EMG) signal analysis. Wavelet transform can also be used to recognize emotions from speech [10] and for person identification based on EMG images [11]. In the case of ECG signal, feature extraction is often based on the analysis of the presence of waves and their shape on a record [12, 13].

## 1.2 ECG classification methods

In [14] authors proposed a support vector machine (SVM) with swarm optimization method for feature selection and model selection. Park et al. [15] utilized generalized k-nearest neighbors (k-NN) called Locally Weighted Regression for ECG classification. Discrete Cosine Transform (DCT) with Random Forest was proposed in [16]. There have been attempts to utilize neural networks for ECG signals classification as well. In [3] authors propose end-to-end training with raw ECG waveforms after heartbeat alignment and segmentation as inputs to a neural network. They use a network with three hidden layers, softmax activation output, and restricted Boltzmann machine pretraining of hidden layers.

More recent approaches involve the utilization of deep learning models in ECG classification task. Kachuee et al. [4] propose a residual network with a 1D convolutional filter directly applied to prepossessed ECG signal. Jun et al. [5] utilize 2D convolution networks. To convert ECG waveform to image each beat of a signal is plotted as a 128 x 128 greyscale image. Deep neural networks require a lot of training data to train. This requirement can be prohibitive in medical domains as labeling high amounts of data by experienced physicians can be costly. For this reason, Weimann et al. explore the possibility of utilizing transfer learning in ECG classification task [17]. They used the Icentia11K dataset [18] for unsupervised pretraining of a Convolutional Neural Network, which later is fine-tuned for the classification task on PhysoiNet/CinC challenge [19]. Several pretraining methods were analyzed, yielding up to 6.57% improvement in F1 score. In [20] authors claim reaching a human-level performance with a deep neural network. This was possible due to the construction of a new big dataset containing 91,232 ECG records from 53,549 patients. Authors of [21] use a subset of available ECG leads to reconstruct information from all channels. This is achieved by unsupervised training of encoder-decoder network with Seq2Seq architecture. Autoencoder training enables for construction of latent space representation that compensates for missing leads. This latent space representation is used to train 1D ResNet for classification.

### 1.3 ECG evaluation schemes

There are two evaluation schemes used for ECG classification namely: a *class-oriented* and a *subject-oriented* [8]. In the *class-oriented* method, data is divided into training and testing subsets, with no consideration of whether signals collected from the same individual are in both train and test sets. This can cause the presence of very similar patterns in both training and testing sets, and it can make evaluation results unreliable. With the *subject-oriented* schema, this kind of problem is omitted by utilizing information about patient identity. As a result, a more robust estimation of true model generalization capabilities can be obtained. Another type of distinction that can be made is the utilization of general or patient-specific data. In most studies, learners are trained on a dataset composed of ECG recording for multiple patients. However, differences in ECG signal properties across patients can be substantial. Therefore some works [22–24] utilize small amount of data labeled for each patient. This data is used to create patient-specific learners that can later detect abnormalities in each heartbeat. Works that employ patient-specific protocol report higher accuracy however, they cannot be directly compared to other articles.

### 1.4 Aims and motivation

The aim of this work is to propose and evaluate a novel feature extraction method for signals. We examine the possibility of combining the independent component analysis with the wavelet transform by modifying the ICA mixing matrix. We hope that including additional information in the ICA feature extraction process will provide a performance boost while keeping an inference time low. These properties can make our solution attractive in certain applications where a high amount of computational power is unavailable. To formalize, we set the following research questions:

1. What is the baseline performance of DWT and ICA applied separately as feature extraction methods for ECG?

2. Is there a benefit from the utilization of DWT as an additional source of information during signal separation in ICA?

3. Are there any alternatives to DWT that can be utilized as auxiliary information sources in ICA?

4. How proposed method compares to other results from the literature?

   We also want to emphasize, that ECG classification performance is not of primary importance, as the main goal of this work is to introduce a novel feature extraction method.

## 2 Materials and methods

In this section, we introduce a novel method for the modification of mixing matrices with the utilization of wavelet transform. We also provide details about signals preprocessing, used dataset, metrics, and experiment setup.

### 2.1 Method

Firstly we define the principal component analysis (PCA) model as:

$$x_{ip} = b_{p1}S_{1i} + b_{p2}S_{2i} + \ldots + b_{pm}S_{mi} = \sum_{j=1}^{m} b_{pj}S_{ji}, \tag{1}$$

where: $x_{ip}$ is value of $p$-th variable for $i$-th feature $p \in \{1, 2, 3, \ldots, m\}$, $i \in \{1, 2, 3, \ldots, n\}$, $S_{ij}$ is value $j$-th principal component for $i$-th feature $j \in \{1, 2, 3, \ldots, m\}$, $b_{pj}$ is principal component

coefficient. The model of principal components is based on matrix operations. It can be represented with matrices as:

$$Z = B \circ S, \tag{2}$$

where: $Z = [Z_1, Z_2, \ldots, Z_m]^T$ is matrix of variables $Z_p = (x_{1p}, x_{2p},.., x_{np})$ with $n$—being number of features. $B = [b_{pj}]_{n \times m}$ is matrix of principal components coefficients and $S(S_1, S_2,.., S_m)^T$ is a matrix of principal components, with $S_j = (s_{j1}, s_{j2}, \ldots, s_{jn})$. Symbol $\circ$ denotes matrix multiplication. Principal components are determined with Hotelling algorithm [25]. Coefficients of principal components are computed in few steps. The coefficient of the first component $S_1$ is determined by maximizing the variance of this component in all variables $w_1$ using the function:

$$W_1 = \sum_{p=1}^{m} b_{p1}^2. \tag{3}$$

Where $W_1$ is principle component.

Maximum is determined by utilization of Lagrange multipliers with bound $\tilde{R} = BB^T$, where $\tilde{R}$ is covariance matrix. In the next step rest of the covariance is computed:

$$\tilde{R}_1 = \tilde{R} - B_1 B_1^T. \tag{4}$$

Where $B_1 = [b_{p1}]$, $p \in \{1, 2, \ldots, m\}$ are values of coefficients for first principal component. Next $\tilde{R}_1$ is substituted into equation $\tilde{R}_1 = BB^T$. Coefficients of the second principal component can be computed by analogy. These steps are repeated until variance in data explained by all components will reach 100%. In principal component analysis correlation coefficient is a measure of independence between principal components [26], which is important while dealing with multivariate normal distribution. In ICA however for estimating independence between variables entropy [27, 28] is used. Entropy $H(X)$ of random variable $X$ determines an average amount for information carried by possible outcomes of $X$ and can be written as:

$$H(X) = -\sum_{i=1}^{k} p_i log_2(p_i), \tag{5}$$

where $p_i$ is probability of outcome $x_i$. Entropy values are nonnegative and zero only when the probability of some outcome is one and all other outcomes probabilities are zero. In the case of the same probability for all outcomes, the entropy value is maximal. To estimate dependence between two variables we utilize mutual information $I(X)$. It is based on the value of entropy for individual variables. Mutual information is calculated as the difference between the entropy of marginal distribution density:

$$I(X) = \sum_{j}(H(X_j) - H(X)). \tag{6}$$

This metric is a modification of Kullback-Leibler divergence for two distributions. In our model we use negentropy $J(X_p)$ for estimating signals dependence. Negentropy is given by the equation:

$$J(X_p) = H(Y_p) - H(X_p), \tag{7}$$

where $Y_p$ is random variable with normal distribution, which variance values is equal to variance of $X_p$ i.e. $VAR(Y_p) = VAR(X_p)$. Negentropy is a measure used in signal processing for independent component analysis. We utilize it as a criterion in blind signal separation. Maximizing the negentropy of the output signals is equivalent to minimizing the mutual information between these signals.

In the proposed extraction method we assume that variables are linear combinations of independent components:

$$\mathbf{X} = \mathbf{ASH} + \mathbf{E},$$ (8)

where:

$\mathbf{X}_{|D| \times n}$—matrix containing learning examples (with $|D|$ being number of learning examples that are currently processed)

$\mathbf{A} = [a_{pj}]_{n \times m}$ is principal component coefficients matrix,

$\mathbf{S} = [\mathbf{S}_1, \mathbf{S}_2, \ldots, \mathbf{S}_m]$ is principal components matrix,

$\mathbf{H}$ is coefficients matrix extracted from signal with wavelet transform from source signal [29],

$\mathbf{E} = [\mathbf{E}_1, \mathbf{E}_2, \ldots, \mathbf{E}_m]$ is noise matrix.

Matrix $\mathbf{A}$ determines principal components coefficients, and its elements are the basis for extracted features. By maximizing negentropy we attempt to obtain principal components that are independent. The process of determining principal component vectors is repeated for each class separately. For each segmented heartbeat, features are computed as principal components coefficients for all classes. Concatenated coefficients are the final feature vector that is utilized for classification. By computing components for each class separately, we hope to obtain distinct representations that will be more unique and therefore will enable easier classification. The number of principal components used for class can vary. An exact number of all extracted features is specified later in this work for each experiment.

Wavelet transform can be written as:

$$F(t) = \sum_{k=1}^{n} c_{j,k} \varphi_{j,k}(t) + \sum_{j=1}^{J} \sum_{k=1}^{n} \Gamma d_{j,k} \Phi_{j,k}(t).$$ (9)

where:

$c_{j,k}$—approximation coefficient for the $j$ scale and $k$ localization,

$d_{j,k}$—detail coefficient,

$\varphi_{j,k}$—time window,

$J$—decomposition value,

$n$—length of the original signal,

$\Phi$—wavelet function

$\Gamma$—low and/or high pass filter;

To allow for utilization of this transform in ICA it needs to be written in matrix form. Therefore we define $\mathbf{L}$ matrix as:

$$\mathbf{L} = \begin{bmatrix} h_0 & h_1 & 0 & 0 & \ldots & \ldots & \ldots \\ 0 & 0 & h_0 & h_1 & \ldots & \ldots & \ldots \\ \ldots & \ldots & \ldots & \ldots & \ldots & \ldots & \ldots \\ \ldots & \ldots & \ldots & \ldots & \ldots & \ldots & \ldots \\ g_0 & g_1 & 0 & 0 & \ldots & \ldots & \ldots \\ 0 & 0 & g_0 & g_1 & \ldots & \ldots & \ldots \\ \ldots & \ldots & \ldots & \ldots & \ldots & \ldots & \ldots \end{bmatrix}$$ (10)

where: $g_0, g_1, \ldots$ is representation of basic function (high-pass filter) $h_0, h_1, \ldots$ is representation of scaling function (low-pass filter). Matrix **L** is the representation of a single heartbeat. To convert this sparse matrix to a more useful form Strang algorithm is utilized [29]. The output of the Strang algorithm for each ECG signal is concatenated into **H** matrix, which can be used in ICA mathematical model. In the proposed method for ECG feature extraction signal Daubechies wavelets were used. They can model a single heartbeat well. From now on we will refer to this kind of wavelets as *db i* where *i* is wavelet number.

## 2.2 Alternative to wavelet transform for ICA modification

In our work, we evaluate the possibility of replacing wavelet function with probability density functions in the process of ECG signal processing. This method is based on calculating values of ECG signals in a time window with the utilization of probability density functions. Wavelet function $\Phi$ in Eq (9) is replaced by one of the functions $F(t)$ proposed below. Six example density functions are provided that can be used to construct $F_i$ vector. Functions with high values properly placed were chosen to obtain good alignment with the R wave.

$$F_1(t) = \frac{\lambda^t}{t!} e^{-\lambda} \tag{11}$$

The first proposed function is Poisson distribution defined on window area $d$, where $\lambda$ is chosen such that the maximum of Poisson distribution cover ECG R wave.

The next utilized density is exponential probability distribution:

$$F_2(t) = \begin{cases} \dfrac{1}{\lambda} e^{-\frac{t}{\lambda}} & for \quad t \geq t_R \\ \dfrac{1}{\lambda} e^{\frac{t}{\lambda}} & for \quad t < t_R \end{cases} \tag{12}$$

Where $t_R$ is a time of wave R appearance in the window of width $d$ and $\lambda$ is coefficient chosen so $t_R$ is aligned with R wave in ECG signal.

The next considered function is t-Student distribution:

$$F_3(t) = \frac{\frac{1}{\sigma\sqrt{2\pi}} exp\left(-\dfrac{(t-\mu)^2}{2\sigma^2}\right)}{\sqrt{z}} \sqrt{r_d} \tag{13}$$

where:

$r_d$—number of signal samples in window $d$

$z$—random variable with chi-square density

$\mu$—mean

$\sigma$—variance

The fourth density is normal distribution:

$$F_4(t) = \frac{1}{\sigma\sqrt{2\pi}} exp\left(-\frac{(t-\mu)^2}{2\sigma^2}\right). \tag{14}$$

where:

$\mu$—mean

$\sigma$—variance

Next is logarithmic distribution:

$$F_5(t) = \frac{1}{\sigma\sqrt{2\pi}t} \, exp\left(-\frac{(ln(t) - \mu)^2}{2\sigma^2}\right). \tag{15}$$

In case of this distribution it is important to remember that first time index $t$ in window cannot be zero.

Last considered function is logistic density:

$$F_6(t) = \frac{1}{1 + (t/\alpha)^{-\beta}} \tag{16}$$

where:

$\alpha = e^\mu$

$\beta = 1/\sigma$

$\mu$—mean

$\sigma$—variance

## 2.3 ECG signal alignment

An important element of the proposed method is heartbeat alignment to fixed window width $d$. In our experiments, windows are selected to match the middle of the window with the R wave. This causes phase alignment of the R wave in the ECG signal with a middle of the wavelet function. Misalignment can introduce noise to extracted features and impact classification performance. To establish the influence of signal alignment on feature extraction quality phase shift coefficient $\alpha$ was introduced. Let $t_1$ be the time index of the window begin, $t_d$ be the time index of the window end, and $d = t_d - t_1$ be the width of the window. The middle of the window is given by the equation:

$$t_{\frac{d}{2}} = t_1 + \left\lfloor \frac{d}{2} \right\rfloor \tag{17}$$

Let $t_R$ be time index of wave R appearance. Phase shift between $\frac{d}{2}$ and $t_R$ is then defined as:

$$\alpha = \frac{\left| t_R - t_{\frac{d}{2}} \right|}{\frac{d}{2}} \tag{18}$$

In Fig 1 phase shift is visualized with sample ECG signals. During preliminary experiments, we observed that signal misalignment and high values of $\alpha$ have detrimental effects on classification quality. There are two possible explanations of this phenomenon. Firstly when lacking proper signal alignment, wavelets do not cover ECG signal correctly. Secondly, misalignment can cause that some features can have a larger set of possible values, as localization of R wave would move from one learning example to the other.

ICA is based on negentropy; therefore, it is important to evaluate how it is affected by phase shift. In Fig 2 negentropy is plotted as function of phase shift $\alpha$. With $\alpha = 0$ negentropy is equal to 0.95, which means that features are extracted well. When $\alpha$ is close to 1 high level of noise is introduced into features. Furthermore, performing classification with $\alpha = 0.0$ and 0.1 cause a recall drop of approximately 14% when using Multi-layer perceptron (MLP) and 15% when

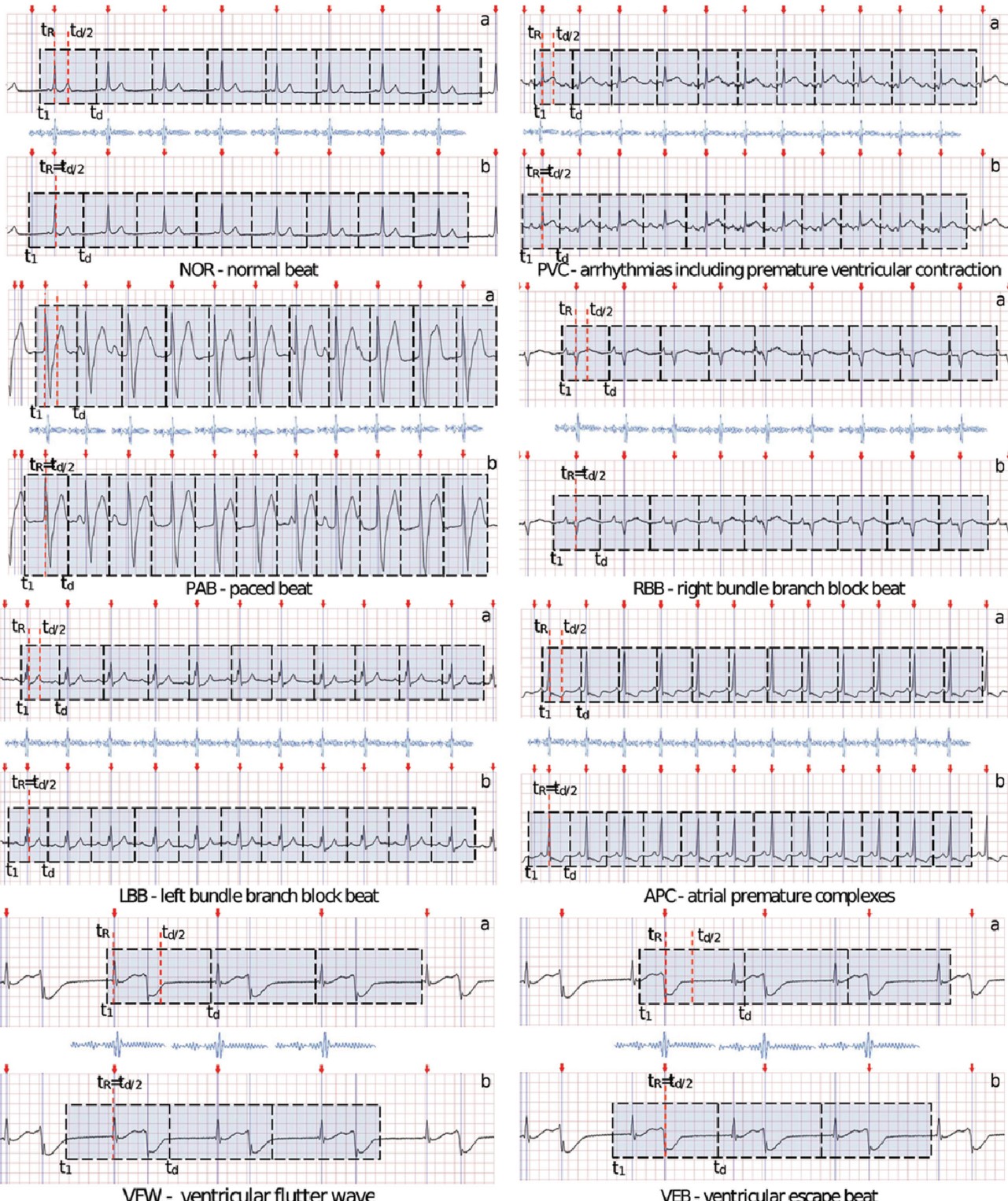

**Fig 1. Visualisation of shift between R peak and middle of heartbeat for different classes of arrhythmia.**

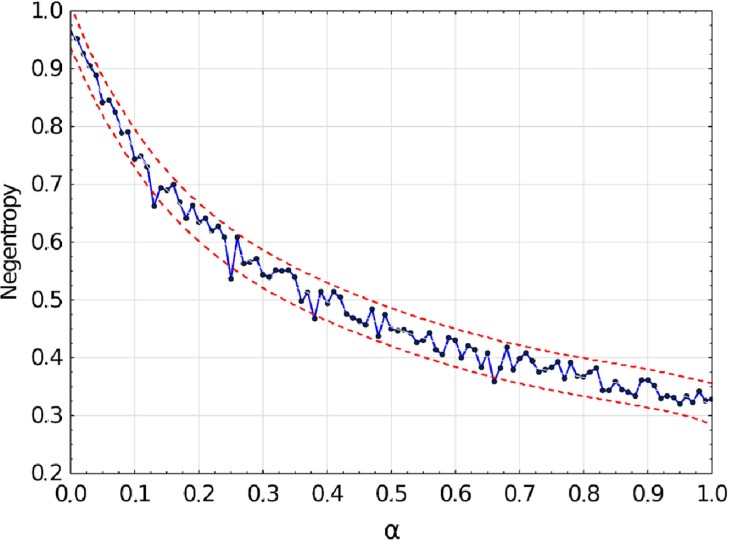

**Fig 2. Negentropy as a function of phase shift $\alpha$ with 95% confidence interval.**

using SVC. In all further experiments, the ECG signal was aligned to contain an R wave in the middle of the window.

## 2.4 Method overview

An overview of the proposed method is provided in Fig 3. The method has three separate steps. Firstly raw ECG signal is preprocessed. Then, segmentation of single heartbeats is performed. R wave is found using Pan-Tompkins algorithm [30]. This is a commonly used algorithm for R wave detection in ECG signals. Each segment is aligned to fixed window size $d$ and to contain an R wave in the middle. The second stage is feature extraction. It starts with covering segmented signal with wavelet function. Next, wavelet transform is computed (9). Due to the utilization of matrix notation in ICA, the calculated wavelet is represented as a matrix of coefficients. In the next step independent components (8) are calculated. Features extracted with modified independent components are stored for training. We want to emphasize that wavelet transform is not the basis of our method. It is used only as an additional source of information about signals for the ICA method. In our work, several base functions for wavelet transform were analyzed.

## 2.5 Used classifiers

Below all classifiers used in the experiments are listed. For each classifier, we provide hyper-parameter sets that were considered for fine-tuning.

- **k-NN**—k Nearest Neighbours [31]

  - number of neighbors: 3, 5, 7

  - metrics: Minkowski, Euclidean, Manhattan

- **SVC**—Support Vector Classification/Support Vector Machine [32]

  - paremeter C: 0.1, 1, 10, 100

  - kernel: linear, rbf, poly, sigmoid

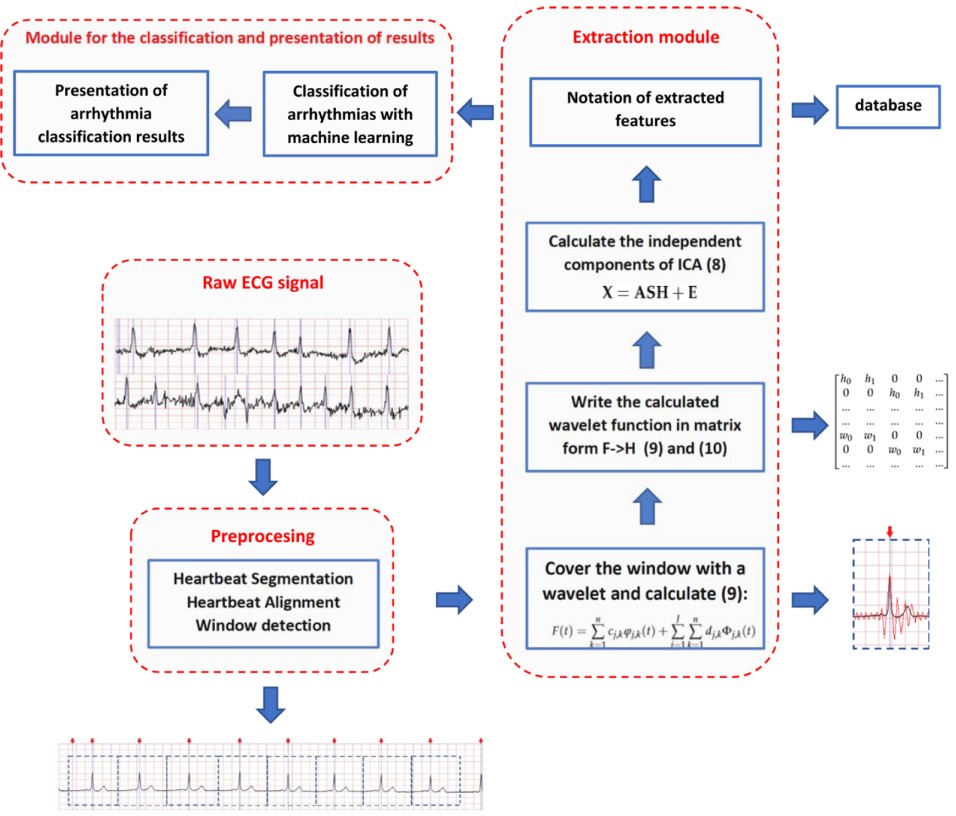

**Fig 3. Concept diagram of proposed method.**

- - gamma: scale, auto
- **CART**—Classification and Regression Trees [33]
  - criterion: gini, entropy
  - splitter: best, random
  - maximum depth: 1, 2, 3, . . ., 10
- **GNB**—Gaussian Naive Bayes—without parameters [34, 35]
- **MLP**—Multi-layer perceptron [36]
  - number of hidden layers: 3,4,5,. . .,10
  - activation function: identity, logistic, entropy, SOS, Tanh, Linear, Softmax, Exponent
  - parameter alpha: 0.00001, 0.0001, 0.001, 0.01, 0.1
  - momentum: 0, 0.2, 0.4, 0.6, 0.8, 1

All hyperparameters were tuned automatically with statistica software.

## 2.6 Metrics

Across experiments standard set of metrics for imbalanced classification was used. For clarity, we provide definitions below. In equations below TP, TN, FP, and FN denote True positive,

True Negative, False Positive and False Negative respectively.

$$Accuracy\ (\%) = 100 \times \frac{TP + TN}{TP + TN + FP + FN} \tag{19}$$

$$Recall\ (\%) = 100 \times \frac{TP}{TP + FN} \tag{20}$$

$$Specifity\ (\%) = 100 \times \frac{TN}{TN + FP} \tag{21}$$

$$Precision\ (\%) = 100 \times \frac{TP}{TP + FP} \tag{22}$$

$$BAC\ (\%) = \frac{Recall\ (\%) + Specifity\ (\%)}{2} \tag{23}$$

Accuracy is not appropriate in an imbalanced classification setting as a model can easily overfit to the majority class and obtain a high value of this metric. Other works in ECG literature report accuracy, so we calculate this metric for completeness, but it will not be used in results analysis. BAC-*score*, Recall, Precision, and Specificity are more suited for imbalanced classification problems. The recall is a percentage of all samples in a dataset with a given class that are correctly classified by the model. Precision is a percentage of model predictions indicating positive examples of a given class that are correct. Specificity is in some sense analogous to recall but is defined on True Negatives instead of True Positives. Depending on the application, one can optimize any of the metrics defined above. In our experiments, we utilize mainly BAC-*score* for comparison between models, as it is often important in the case of medical applications. In this field keeping a low number of false negatives is vital as each positive case can be associated with some disease, require the intervention of doctors, further examination, and diagnosis. Not detecting positive cases by the model can have detrimental effects on a patient's health or life. All metric values reported in this work were obtained with five fold cross-validation. For evaluation of our method Bac-*score* was used. For comparison to other results from the literature, all of the above metrics were reported.

## 2.7 Dataset

PhysioNet [1] MIT-BIH Arrhythmia Database [2] was utilized for experiments. The database has been maintained since 1980, and it aimed to collect and standardize ECG signals for arrhythmia classification. It includes 24 hours of signals collected from 47 patients, including 25 men aged 32-89 years and 22 women aged 23-89 years. Each patient is represented as one record (except for one with two records) numbered from 100 to 124 and from 200 to 234. Due to different medical conditions of patients, the leads used to measure the heart's activity are not uniform. All records contain lead II (referred to also as MLII).

ANSI/AAMI EC57 contains requirements for ECG classification algorithms, recommendations for individual records, and a definition of the arrhythmia classes. For records it is advised to discard patients with pacemakers under numbers 102, 104, 107, and 217. The division of heart activation types into classes recommended by the standard is presented in Table 1. The total number of all heartbeats is 101464. Due to the low number of learning examples and the ambiguity in defining the features, the Q activation was not taken into consideration in this study.

**Table 1. Labels used in the MIT-BIH database with number of learning examples assigned to each class.**

| Class name | #learning examples |
| --- | --- |
| Normal(N) | 90632 |
| Vestibular ectopic beats (S) | 2779 |
| Ventricular ectopic beats (V) | 7129 |
| Fusion strikes (F) | 803 |
| Undefined (Q) | 15 |

From the classification point of view, it is important to divide the data set into training and testing sets. During the dataset split, we utilized information about patient identity to avoid leaking of the patterns that can be observed for one subject from training to test set. To correctly divide the dataset, we also need to take into consideration the number of learning examples assigned to the classes. The proportion of data in train and test split is defined by stratified 5-fold cross-validation [37].

## 2.8 Signal preprocessing

The raw ECG signal contains a lot of noise and artifacts that must be filtered out to obtain better and more reliable classification results. We can distinguish two types of interference with an ECG signal. The first type is the baseline wander—the disruption caused by the patient's breathing, movements, and electrode displacement. The second type of interference is ECG convolution with a signal with a frequency of 60Hz coming from the AC power supply of the electrocardiograph.

A low-pass filter with 5-15 Hz was utilized to eliminate noise generated by muscle tremors, electricity power supply (50/60 Hz), and the influence of T wave and floating isoline. Filter poles were localized on the unit circle to cancel out zeros of Laurent transform polynomial [30]. The result of filtration was subtracted from the original ECG waveform to obtain a signal without this kind of noise. In Fig 4 filtered ECG signal is presented for different classes that are present in the MIT-BIH dataset.

## 2.9 Experiments setup

Experimental evaluation was divided into several parts. Firstly features were extracted with wavelet transform with ten types of wavelets db1-10. The main goal of the first experiment is to obtain an assessment of classification performance when ICA and wavelet feature extraction is performed separately. Wavelets used in this experiment are presented in the Fig 5. The next proposed extraction method was evaluated. Same classifiers and types of wavelets as in the first experiment were employed. The goal of these two experiments was to examine if the proposed method improves classification scores compared to the utilization of wavelet transformation and ICA separately. To compare baseline to proposed method Wilcoxon test was performed for each type of wavelet and classifier separately with significance level 0.05. The null hypothesis is that these two methods performance is not significantly different from each other. The third experiment verifies if alternative ICA mixing matrix modification methods provide better classification performance than results from the previous experiments.

In the first experiment, 26 features were extracted with wavelets and 23 with ICA. In the second experiment, 21 features were used and 12 features in the third experiment. In all cases, features were extracted with 200 max iteration, coefficient $\alpha = 0$ with neg-entropy log-cosh.

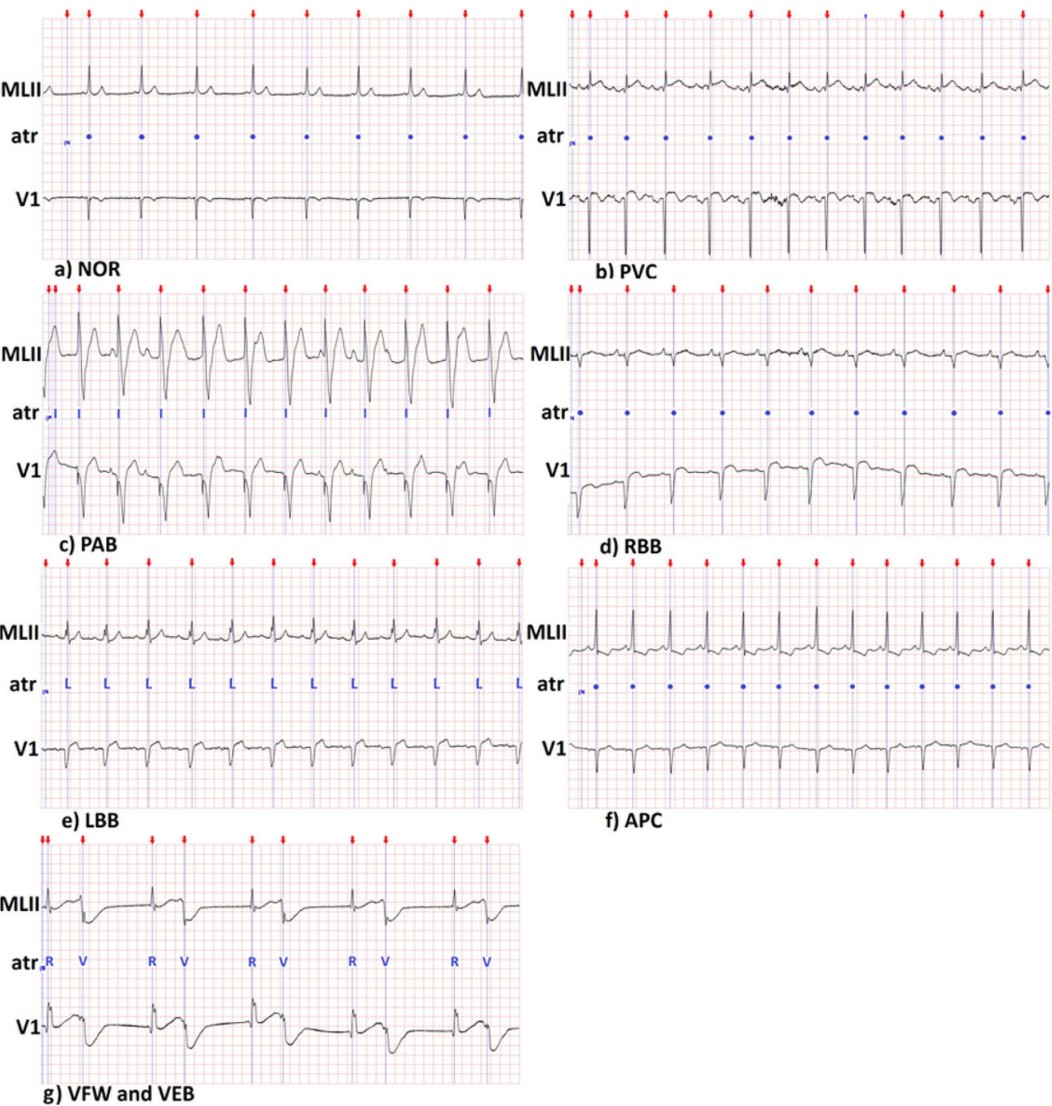

**Fig 4. Example ECG signal after applying filtration and alignment of QRS complex.** Following type of beats were presented: a) NOR—normal beat, b) PVC—arrhythmias including premature ventricular contraction, c) PAB—paced beat, d) RBB—right bundle branch block beat, e) LBB—left bundle branch block beat, f) APC—atrial premature complexes, g) VFW—ventricular flutter wave, VEB—ventricular escape beat.

## 3 Results

This section contains the results of the experimental evaluation according to the experiment setup provided above.

### 3.1 Classification quality for wavelet transform and ICA applied separately

In this part of the experiments, we provide a baseline that will serve for comparison later. Results are presented in Fig 6. Based on these values we can conclude that the best classification scores are obtained for the MLP network. The highest metrics values for this kind of network were obtained for each type of wavelet and ICA. In the case of ICA BAC-*score* of MLP network is 92.23%. Of all wavelet functions, best BAC-*score* is obtained for db6: 91.95%. SVC classifier for ICA method obtained 91.00% BAC-*score*, while for db6 wavelet SVC BAC-*score*

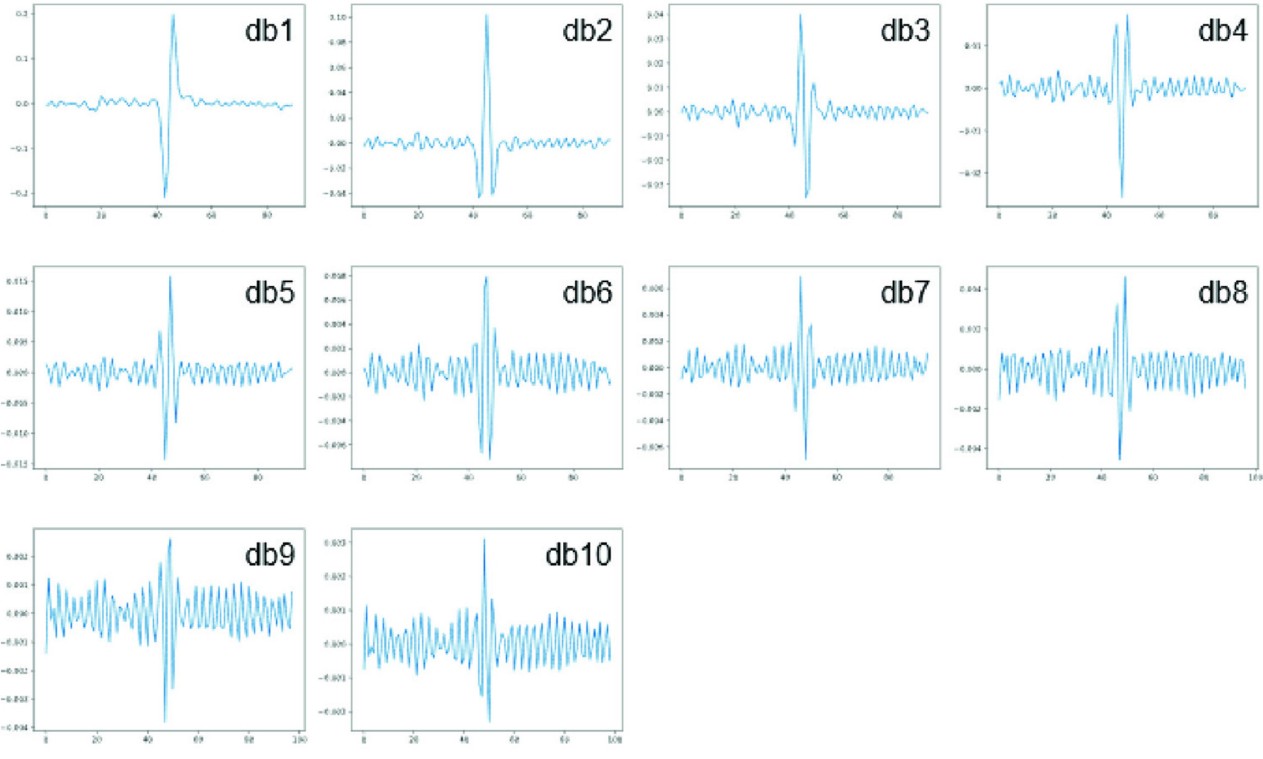

**Fig 5. Examples of Daubechies 1-10 wavelets used in experiments 1-3.**

was equal to 90.94%. The lowest classification scores were obtained for the Naive Bayes classifier with values ranging from 79.83% to 85.66%.

## 3.2 Classification quality of proposed method

In this section proposed feature extraction method was evaluated. Obtained metrics are provided in Fig 7. After the application of the proposed method similar to the first experiment 5-layers MLP obtained the best results. With db6 it obtained 95.79% BAC-*score*. SVC scored 94.27% BAC-*score* with the same wavelet.

Comparing obtained results from previous experiments, we note that after applying the proposed method increase in metric values is in the range from 3.29% to 3.86%. The average increase is 3.68%. In both experiments, the best results were obtained for the db6 wavelet. In Table 2 p-values for Wilcoxon test are presented. The presented values of significance p for the Wilcoxon test relate to the comparison of the classification quality of the proposed ICA method with the db6 wavelet with the quality results obtained after extraction with wavelets only. With a statistical significance level of 0.05, we reject the null hypothesis and conclude, that results are significantly different for all types of wavelets. Therefore, we can argue that our feature extraction method indeed provides additional information that impacts the quality of extraction in an independent analysis model.

## 3.3 Classification quality when utilizing different function for ICA mixing matrix modification

We test how ICA mixing matrix modifications based on probability distribution functions affect the quality of classification. Obtained results were also compared with the best type of

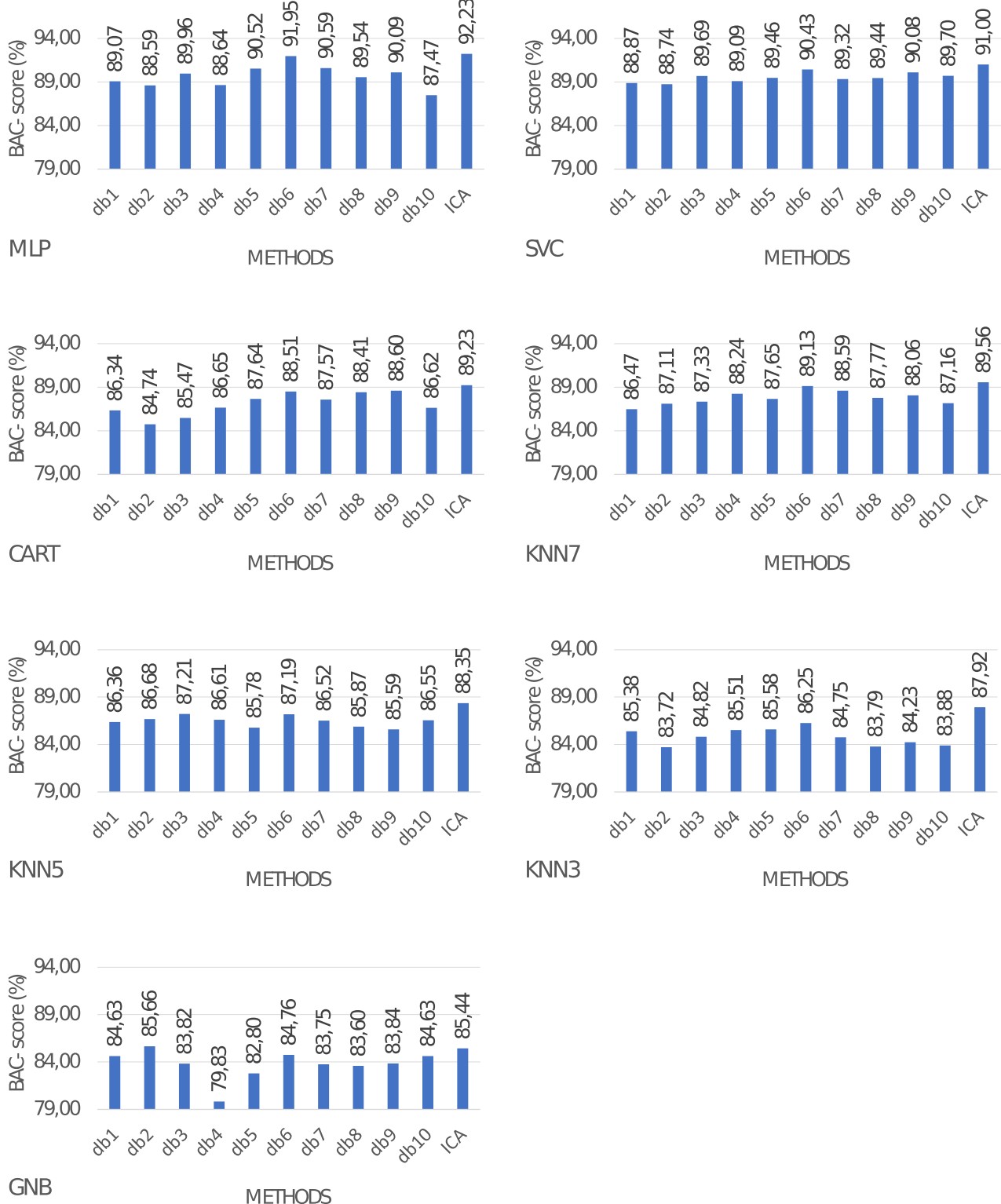

**Fig 6. Classification performance when the wavelet transform or ICA are applied separately.**

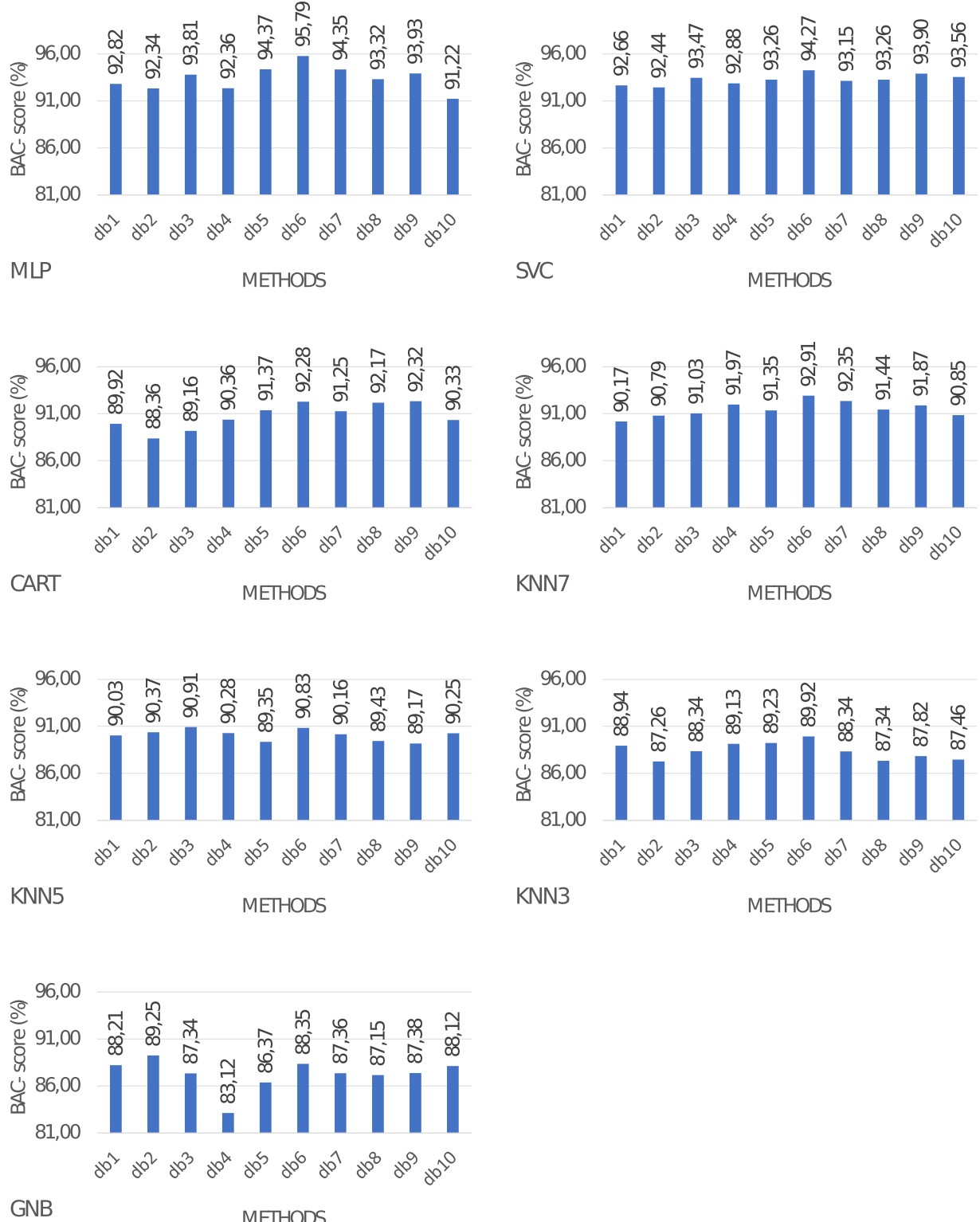

**Fig 7. Classification performance when utilizing different wavelet functions for modification of mixing matrix in ICA.**

**Table 2. P-values obtained with Wilcoxon test comparing BAC-*score* from expierment 1 and 2.**

| METHODS | SVC | KNN7 | KNN5 | KNN3 | MLP | CART | GNB |
|---|---|---|---|---|---|---|---|
| db1 | 0.019 | 0.016 | 0.019 | 0.018 | 0.021 | 0.02 | 0.019 |
| db2 | 0.02 | 0.013 | 0.019 | 0.01 | 0.013 | 0.014 | 0.015 |
| db3 | 0.013 | 0.016 | 0.011 | 0.01 | 0.017 | 0.019 | 0.01 |
| db4 | 0.016 | 0.018 | 0.012 | 0.014 | 0.017 | 0.018 | 0.015 |
| db5 | 0.016 | 0.017 | 0.01 | 0.019 | 0.014 | 0.015 | 0.017 |
| db6 | 0.015 | 0.017 | 0.011 | 0.018 | 0.013 | 0.019 | 0.011 |
| db7 | 0.018 | 0.016 | 0.013 | 0.014 | 0.014 | 0.013 | 0.011 |
| db8 | 0.019 | 0.015 | 0.014 | 0.011 | 0.013 | 0.013 | 0.01 |
| db9 | 0.017 | 0.019 | 0.018 | 0.013 | 0.01 | 0.018 | 0.015 |
| db10 | 0.012 | 0.02 | 0.018 | 0.017 | 0.015 | 0.014 | 0.011 |

wavelet from previous experiments. Metrics are presented in Fig 8. From these results, we can conclude that classification performance is worse compared to the utilization of db wavelets. The best BAC-*score* observed so far is 95.79% for MLP and 94.27% for SVM. For other functions of the mixing matrix, modification BAC-*score*l varies from 77.37% to 89.21%. Differences between these methods and ours are in rage 6.60%-10.97%.

Based on conducted experiments for further experiments, we utilize MLP classifier with wavelet db6 type. It is the model that obtained the best results, as demonstrated by our experiments.

## 4 Discussion

Experiments described so far were conducted with various machine learning algorithms i.e. SVC support vector machines, K of the nearest neighbors for K = 3,5,7, MLP neural network, CART classification tree, and GNB naive Bayes classifier. Firstly it was demonstrated that wavelet transformation could score between 79.83% and 91.95% BAC-*score* and ICA can score between 85.44%, and 92.23%. These results were a baseline for further comparison, and they provide an answer for research question 1. After applying our method (i.e. actualization of ICA matrix with wavelet transform matrices) an increase in BAC-*score* up to 11% was obtained. We want to emphasize the importance of signal alignment and matching of wavelet function with R wave in ECG signal. Even a small shift in phase between R wave of ECG and middle of window can have a detrimental impact on classification performance. In our experiments, MLP was the best performing classifier. This answers research question 2. When utilizing density probability function as an additional source of information for ICA instead of DWT obtained BAC-*score* was in 77.37, 89.21 interval. Obtained metrics for alternative functions were lower for all classifiers than for the db6 wavelet. This answers research question 3.

In years 2006-2012 first approaches that combined DWT with ICA were proposed [8, 38]. These works concentrated on EEG signals. Wavelet transform was utilized for signal preprocessing and denoising. Next, preprocessed signals were separated into independent sources with ICA [38]. So in principle, one can describe these methods as conducting ICA in the DWT domain. Our method actualizes the matrix of independent components with information obtained with wavelet transforms. Due to this fact, we were able to obtain better results compared to other feature extraction methods [39, 40]. In [8] authors also utilize wavelet transforms with ICA for feature extraction however, they use ICA and wavelet as separate components of feature vectors, while our method combines these two algorithms.

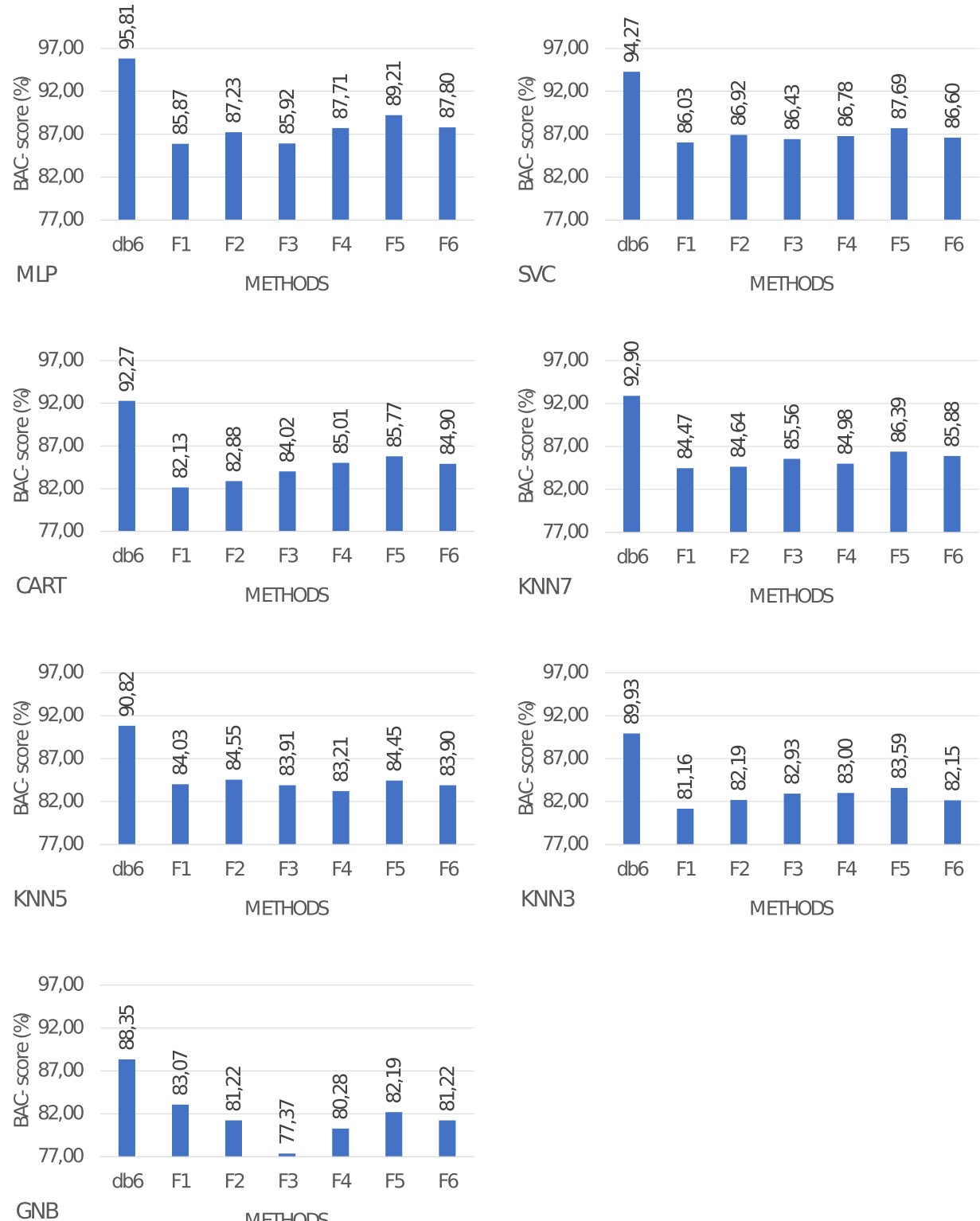

**Fig 8. Classification metrics for various functions for the mixing matrix creation in ICA.**

**Table 3. Best MLP networks used for heartbeat classification task.**

| Id | Network name | Accuracy | Learning algorithm | Loss function | Activation (hidden) | (Activation (output) |
|----|----|----|----|----|----|----|
| 1 | MLP 5 | 95.58 | BFGS | Entropy | Exponent | Softmax |
| 2 | MLP 6 | 93.74 | BFGS | Entropy | Tanh | Softmax |
| 3 | MLP 3 | 93.24 | BFGS | SOS | Exponent | Tanh |
| 4 | MLP 3 | 92.99 | BFGS | Entropy | Tanh | Softmax |
| 5 | MLP 5 | 92.74 | BFGS | Entropy | Linear | Softmax |
| 6 | MLP 7 | 92.71 | BFGS | Entropy | Linear | Softmax |
| 7 | MLP 3 | 92.49 | BFGS | Entropy | Exponent | Softmax |
| 8 | MLP 6 | 92.47 | BFGS | SOS | Linear | Exponent |
| 9 | MLP 7 | 92.37 | BFGS | SOS | Linear | Exponent |
| 10 | MLP 5 | 92.24 | BFGS | Entropy | Linear | Softmax |

## 4.1 Evaluation details

The proposed method was compared to ECG beat classification or feature extraction algorithms from the literature. Other works utilize seven classes instead of four. For this reason, new class definitions were adopted for comparison. Primary experiments utilize different class definitions due to the fact that the main goal of this paper is to introduce a novel signal feature extraction method. Therefore, obtaining the highest metrics for the ECG beat classification task is not of primary importance. Through the classification task, we only evaluate the usefulness of the feature extraction method. Reevaluation with the best-performing model was performed. As previously, results were obtained with 10-fold stratified cross-validation.

To perform evaluation, we select the best performing model from experiments and conduct a hyperparameter search. Wavelet db6 was used for ICA modification with 21 features extracted. MLP network was selected with the following hyperparameters considered: with 3-20 hidden layers, loss functions: MSE, Cross entropy, and following activation functions for hidden and output layers: linear, sigmoid, tanh, and exponent. Results are provided in Table 3. For further experiments, we have selected the MLP neural network with the softmax output activation function, BFGS neural network learning algorithm (Broyden Fletcher Goldfarb Shanno), and the error function in the form of a sum of squares. This model will be used for comparison with other works.

## 4.2 Comparison to other works

In Table 4 metrics obtained for the proposed method, other extraction methods and selected works from ECG beat classification literature are presented.

The results marked in the methods as "This Paper" concern the authors' use of various extraction methods known from the literature for the extraction of the features of the ECG signal. For various methods, i.e. main components of PCA, factor rotation according to CCPCA class centroids, optimization of rotation angle using GPCA gradients and non-linear kernel function transformation of KPCA, the quality of classification was worse than in the case of the proposed method. Combining different extraction techniques allows, as we can see from the research, to increase the quality of arrhythmia classification. Therefore, it is worth paying attention to the CCPCA and GPCA extraction methods. Application of factor rotation by class centroids, i.e. types of arrhythmias, increases the classification quality by more than 2% compared to the classical PCA method.

Analyzing these results we can draw a conclusion that solutions based on PCA obtain slightly lower accuracy compared to our method. Combining PCA and wavelet transform also

**Table 4. Comparison of our methods with other results reported in literature.**

| Methods | Features | Classifier | Acc | Re | P$_R$ | Sp |
|---|---|---|---|---|---|---|
| Proposed | ICA + Wavelet | MLP | 95.58 | 95.42 | 95.92 | 95.82 |
| This Paper | ICA [41] | MLP | 92.23 | 92.12 | 92.18 | 92.15 |
| This Paper | PCA [42] | MLP | 89.32 | 88.82 | 89.11 | 87.64 |
| This Paper | CCPCA [43] | SVM | 91.45 | 90.53 | 91.08 | 91.12 |
| This Paper | GPCA [44] | SVM | 91.58 | 90.76 | 91.14 | 91.18 |
| This Paper | KPCA [45] | MLP | 90.23 | 90.66 | 89.89 | 89.79 |
| Martis et al. [40] | PCA + Bispectrum | SVM | 93.48 | - | - | - |
| Martis et al. [40] | PCA + Cumulant | NN | 94.52 | - | - | - |
| Martis et al. [40] | PCA + Cumulant + DWT | SVM | 93.76 | - | - | - |
| Chazal et al. [39] | Morphology and heartbeat interval | LD | 85.90 | - | - | - |
| Melgani et al. [46] | PSO + SVM | SVM | 91.67 | 93,83 | - | 90,49 |
| Kiranyaz et al. [47] | 1D CNN | 1D CNN | 96.40 | 68.,68 | 79.2 | 99,50 |
| Jun et al. [5] | 2D CNN | 2D CNN | 99.05 | 96.85 | 98.55 | 99,50 |
| Dutta et al. [48] | SVM | SVM | 95.82 | 86.16 | 97.01 | 99,17 |
| Kumar et al. [16] | DCT | RF | 92.16 | - | - | - |

does not provide significant improvement. We argue that PCA, GPCA, CCPCA, KPCA assigns similar amplitudes of signal in some time proximity to the same principal component, which can explain lower classification performance. In signal fragments with significant variance in amplitude across multiple training samples, PCA can be beneficial. This problem does not occur when using ICA.

Results from [49] are better compared to our experiments with DWT and ICA applied separately presented in Fig 6, and at the same time are worse compared to the proposed method. In [40] authors propose the utilization of ICA and wavelet transform separately with SVM classifier. Metrics obtained for this solution are also worse compared to our method, proving that combining ICA and wavelet transform is beneficial for ECG beat classification performance. Both 1D and 2D convolutional neural networks obtained superior classification performance compared to our method. This answers research question 4.

From provided results, we can conclude that the best metrics are obtained by convolutional neural networks. This broad group of models is a class of its own when considering performance. For this reason, we provide a more detailed comparison with this type of network. We select [5], as this work provides detailed experiments. Three models were selected from this work comparison: AlexNet [50], VGGNet [51] and custom architecture proposed in [5]. Our results are compared to selected convolutional networks in Table 5. The best neural network obtained from the experiment results from the Table 3 was used for comparison. For

**Table 5. Best MLP network for heartbeat classification compared to other results from literature.**

| Method | Grade | Acc(%) | Sp(%) | Re(%) | Pr(%) |
|---|---|---|---|---|---|
| Proposed | | 95.58 | 95.82 | 95.42 | 95.92 |
| Jun et all [5] | Native | 98.90 | 99.64 | 97.20 | 98.63 |
| Jun et all [5] | Augmented | 99.05 | 99.57 | 97.85 | 98.85 |
| AlexNet [50] | Native | 98.81 | 99.68 | 96.81 | 98.63 |
| AlexNet [50] | Augmented | 98.85 | 99.62 | 97.08 | 98.59 |
| VGGNet [51] | Native | 98.77 | 99.43 | 97.26 | 98.08 |
| VGGNet [51] | Augmented | 98.63 | 99.37 | 96.93 | 97.86 |

comparison, cases with data augmentation increasing the training data set and with native content of ECG images were taken into account.

Obtained classification metrics for the proposed method are approximately 3-4% worse compared to CNNs. For more detailed analysis, performance for separate classes needs to be considered. Analyzing results in Fig 9 we conclude that the MLP network for arrhythmia types: PVC, PAB, RBB, LBB obtained worse performance, but it can be competitive with CNNs for arrhythmia types APC, VFW, and VEB. We presume that the advantage of results from [5] is connected to the number of learning examples available for each of the classes. Convolutional neural networks, when evaluated on classes with the lower amount of available data obtain comparable results. Therefore, we argue that our method can be on a par with convolutional neural networks in low data regimes.

### 4.3 Computational requirements

Another important consideration is the computational requirements of each method. At test time, our method utilizes pre-computed matrices of independent components. Therefore, no computation of DWT is needed. For this reason number of operations required for computing the full feature vector of an unknown ECG heartbeat can be estimated as:

$$nC\max_c (q_c) \tag{24}$$

where $n$ is the number of samples in one heartbeat, $C$ is the number of classes, and $q_c$ is the number of independent components for class $c$. There is also a cost of classification, that is inherent to all methods and can vary depending on the type of classification algorithms used. Next, we compare the number of operations required in our approach to the convolutional neural network, as these are currently best-performing models. The number of operations required for 1D convolution layer inference with padding added to keep the size of the feature map the same can be written as:

$$Ch_{In}Ch_{Out}kD_f \tag{25}$$

where $Ch_{In}$ is number of input channels, $Ch_{Out}$ is number of output channels, $D_f$ is feature map size, and $k$ is filter size. Please note that in the above expression cost of applying bias was omitted for simplicity. The method proposed in this article will require fewer operations than a single convolutional layer if the following condition is met:

$$nC \max_c (q_c) < Ch_{In}Ch_{Out}kD_f \tag{26}$$

Length of signal $n$ is usually a few hundred samples (162 in case of our experiments) and can be adjusted by subsampling when needed. The number of independent components extracted per class is a hyperparameter that can be tuned. Earlier in this work, we reported the number of all features (i.e. number of independent components for all classes) as close to 20, therefore for simplicity, it can be assumed that $\max_c(q_c) = 20$. The number of classes depends on the task and used dataset. In our experiments, it was 4. For comparison to others, seven classes were used. Now we analyze the right side of the inequality. With increasing neural network depth usually $Ch_{In} Ch_{Out}$ grows larger and $D_f$ gets smaller. The number of input and out channels can change depending on architecture design. Usually they obtain high values in last layers such as 128, 256 or 512 [5, 51]. Filters are usually small (commonly used sizes are 3 or 5 [51]). For this reason, we argue that the condition given above can be easily met when the number of classes $C$ is low and the processed signal $n$ is sufficiently short. Therefore we conclude that with moderate assumptions satisfied proposed feature extraction method can

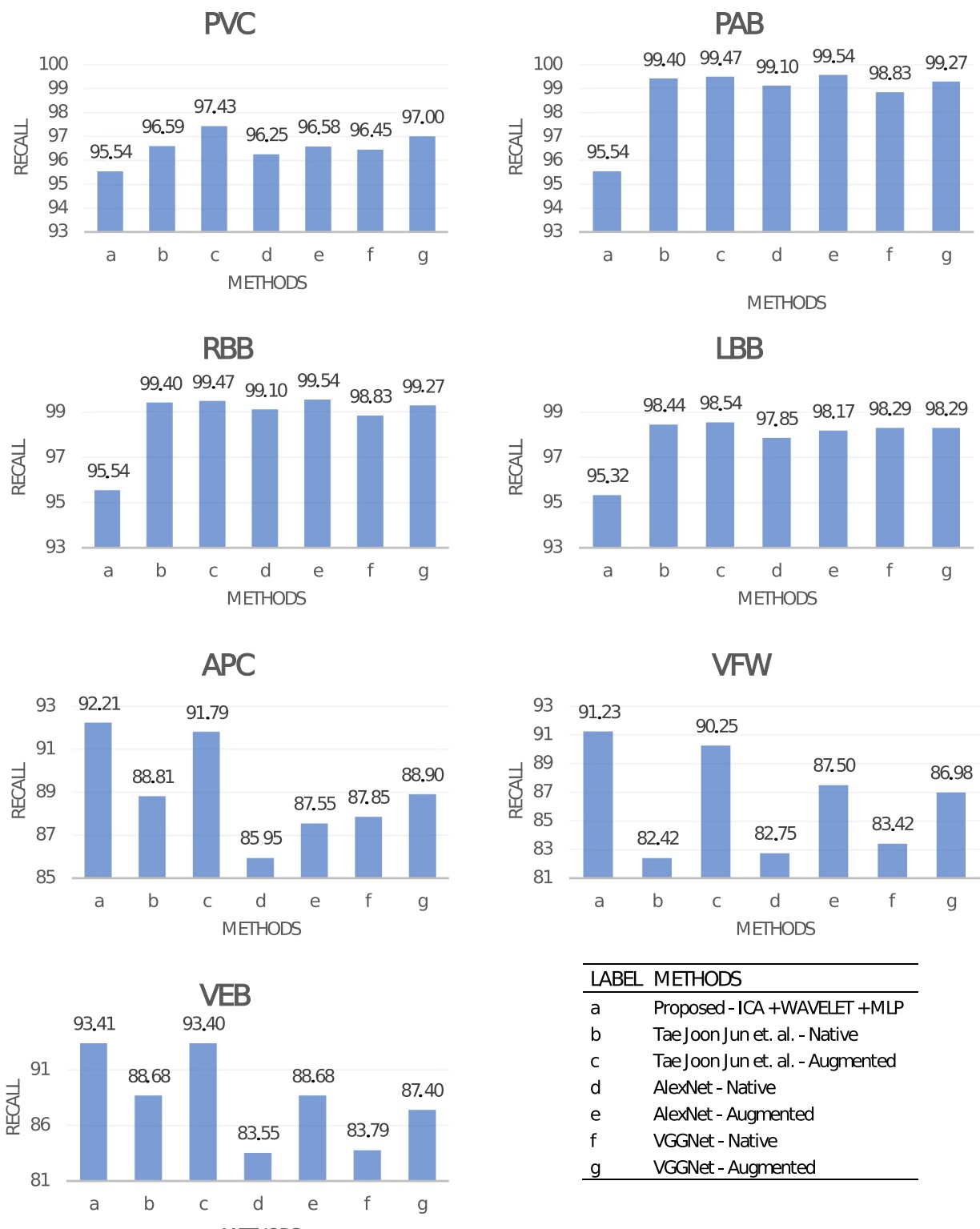

**Fig 9. Comparison of Recall obtained for our method and CNNs from [5] for selected type of beats.** Number of heartbeats for each type is equal to: PVC—7130, PAB—7028. RBB—7259, LBB—8075, APC—2546, VFW—472, VEB—106. For APC, VFW and VEB beats with lower amount of learning examples available our method provided better results than convolutional neural networks.

require fewer operations than a single 1D convolution layer at the top of the network. Neural networks usually contain multiple layers and other operations such as pooling, nonlinear activations, and batch normalization. Layers with 2d convolutions require more operations. Lower compute requirements can allow for easier method deployment in practical scenarios.

## 5 Conclusions

This paper proposes a new approach to the feature extraction task using the independent component analysis, where the wavelet transform is used for the construction of the mixing matrix. This approach is particularly applicable to signals whose values are represented in the time domain. Metrics obtained in experiments were confronted with other results that can be found in the literature. Our method compares favorably to several other works. Convolutional neural networks can obtain better performance, however for classes with a low amount of samples our method can obtain comparable results. Obtaining a large dataset required by deep neural networks can be costly when at least one medical doctor must label each training sample. We argue that our method can be useful in these cases. Also, as shown by our analysis proposed algorithm requires fewer computations at test time. This can enable more broad applications with edge computing or embedded devices.

A new method of extracting the features of the ECG signal was developed, combining the independent component analysis with the wavelet transform. The essence of this approach is that the wavelet transformer modifies knowledge in the form of a matrix of independent components. The obtained results confirmed that such a solution is competitive and gives better classification qualities compared to known extraction methods. The new method also opens up possibilities for analyzing other biomedical signals and more. Furthermore, the data processing speed makes it applicable for quick analysis of the ECG signal, where the signal is taken from the EKG device. However, in the case of convolutional networks, real-time processing may turn out to be too costly and thus difficult.

Future work can include developing feature extraction methods that are more robust to signal misalignment. As the presented algorithm can be utilized for any signal defined in the time domain, adaptation to other applications can be considered. Features extracted with this method can also be clustered to detect a group of anomalies in signals automatically. Some machine learning algorithms require more data to perform well. This property is called data efficiency. Due to the high cost of obtaining large-scale labeled datasets in the medical domain interesting direction of research can be a principled evaluation of data efficiency for existing ECG classification approaches.

## Author Contributions

**Writing – original draft:** Mariusz Topolski.

**Writing – review & editing:** Mariusz Topolski, Jędrzej Kozal.

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
