## [Decision Letter · Decision Letter 0]

28 Sep 2021

PONE-D-21-24171Novel Feature Extraction Method for Signal Analysis based on Independent Component Analysis and Wavelet TransformPLOS ONE

Dear Dr. Topolski,

Thank you for submitting your manuscript to PLOS ONE. After careful consideration, we feel that it has merit but does not fully meet PLOS ONE’s publication criteria as it currently stands. Therefore, we invite you to submit a revised version of the manuscript that addresses the points raised during the review process.

We look forward to receiving your revised manuscript.

Kind regards,

Mahmoud Al Ahmad, PhD

Academic Editor

PLOS ONE

Journal Requirements:

"his work was supported by the Polish National Science Center, grant No. 2017/27/ B/ST6/01325"

"This work was supported by the Polish National Science Center, grant No. 2017/27/\\\\B/ST6/01325"

4. We note that Figure 3 in your submission contain copyrighted images. All PLOS content is published under the Creative Commons Attribution License (CC BY 4.0), which means that the manuscript, images, and Supporting Information files will be freely available online, and any third party is permitted to access, download, copy, distribute, and use these materials in any way, even commercially, with proper attribution. For more information, see our copyright guidelines: http://journals.plos.org/plosone/s/licenses-and-copyright.

1. You may seek permission from the original copyright holder of Figure 3 to publish the content specifically under the CC BY 4.0 license. 

2. If you are unable to obtain permission from the original copyright holder to publish these figures under the CC BY 4.0 license or if the copyright holder’s requirements are incompatible with the CC BY 4.0 license, please either i) remove the figure or ii) supply a replacement figure that complies with the CC BY 4.0 license. Please check copyright information on all replacement figures and update the figure caption with source information. If applicable, please specify in the figure caption text when a figure is similar but not identical to the original image and is therefore for illustrative purposes only

Reviewers' comments:

Reviewer's Responses to Questions

**Comments to the Author**

1. Is the manuscript technically sound, and do the data support the conclusions?

Reviewer #1: Yes

Reviewer #2: Yes

2. Has the statistical analysis been performed appropriately and rigorously? 

Reviewer #1: Yes

Reviewer #2: Yes

3. Have the authors made all data underlying the findings in their manuscript fully available?

Reviewer #1: Yes

Reviewer #2: Yes

4. Is the manuscript presented in an intelligible fashion and written in standard English?

Reviewer #1: No

Reviewer #2: Yes

5. Review Comments to the Author

Reviewer #1: The authors propose a novel method for signal feature extraction that combines analysis of the signal waveform and separation of its sources. A wavelet transform is used to modify a mixing matric in the independent component analysis. The approach was validated using an ECG heartbeat classification, against 4 target classes; normal, vestibular ectopic beats and ventricular ectopic beats and fusion strikes. A 95.79% recall was obtained and the authors claim the method performance is close to that of convolutional neural networks for classes with limited number of learning examples and can be easily deployed in environments with limited computing power. Interesting research questions.

1. Include references for the following;

Line 11 “… well-established datasets and evaluation procedures”

Line 12”… well-performing algorithms developed specifically for this field.”

Line126 “In principal component analysis correlation coefficient…”

Line 314 “The proportion of data in train 314 and test split is defined by stratified 5-fold cross validation.”

Line 405 “In years 2006-2007 first approaches that combined DWT with ICA were proposed.”

Line 410 “Due to this fact, we were able to 410 obtain better results compared to other feature extraction methods.”

2. Several abbreviations should be defined before use; ECG, EMG, TP, TN, FN, FP, MLP

3. Certain sentences require commas to read better. Grammatical errors should be corrected; line 27-28, 58-60, 67-68, 73-75,90-91,138-139, 311-312, 469 (This is not an exhaustive list. Authors are advised to re-read the manuscript)

4. Section 1.1 and section 1.2 are both discussed as feature extraction methods. Section 1.1 should therefore be renamed appropriately perhaps as Independent Component Analysis

5. Consistency should be maintained. Equation (17) D is undefined. Does it refer to the width of the window which was defined as d in the text? Ensure all variables used in equations are defined e.g Df in eqn 24 & 25.

6. Justification for using negentropy should be included.

7. Line 276 ci?

8. Table 5 and 6 should be swapped to correspond with the order in which they are discussed in the text to reduce confusion.

9. Table headings could be more descriptive e.g “Table 5 Best MLP network for heart beat classification compared to selected convolutional networks reported in the literature”

10. Authors argue that in low data regimes the proposed method is comparable to convolutional neural networks, this could be included in the comparison tables for ease of comparison.

11. Justification for some choice of algorithms should be included e.g why stratified 5-fold cross validation, Pan-Tompkins algorithm etc… were these selected because they are commonly used in the literature? Could the technique benefit from other choices?

Reviewer #2: The contribution of this paper is exciting and original. The proposed method seems to be valid for signal feature extraction. The procedure using signal processing can successfully lead to separation between the signal waveform from its resources. Experiments were carried out with an MIT-BIH database to illustrate the four classifications(Normal, Vestibular ectopic beats, Ventricular ectopic beats, and Fusion strikes). The derivations, experimental setup, and discussion section are clear to the readers. I think this paper will exhibit a better understanding of ECG heartbeat classification. Therefore, I recommend accepting this manuscript as it is with some polishing comments:

C1. The abstract shall be rewritten to reflect all results yielded. C2. The Introduction could be shorter with more concentration on state of the art and the advantage of the proposed methodology. C3: In any case, profound grammatical revisions are needed to render the document useful for readers. C4: The article should have clear arguments regarding the proposed method's rationale and motivations and explicit discussions of the advances versus the prior art.

6. PLOS authors have the option to publish the peer review history of their article (what does this mean?). If published, this will include your full peer review and any attached files.

Reviewer #1: No

Reviewer #2: No

---

## [Author Response · Author response to Decision Letter 0]

23 Oct 2021

Reviewer #1 

Thank you very much for your substantive comments allowing us to improve our article. 

RV: Reviewer 

AA: Author Answer

RV: 1. Include references for the following; 

Line 11 “… well-established datasets and evaluation procedures”

AA: Proper references were added.

RV: 1. Include references for the following;

 Line 12”… well-performing algorithms developed specifically for this field.”

AA: Proper references were added.

RV: 1. Include references for the following;

 Line126 “In principal component analysis correlation coefficient…”

AA: Proper reference was included.

RV: 1. Include references for the following; 

Line 314 “The proportion of data in train 314 and test split is defined by stratified 5-fold cross validation.”

AA: Proper reference was included.

RV: 1. Include references for the following;

 Line 405 “In years 2006-2007 first approaches that combined DWT with ICA were proposed.”

AA: This line was removed due to remarks provided by second reviewer.

RV: 1. Include references for the following; 

Line 410 “Due to this fact, we were able to 410 obtain better results compared to other feature extraction methods.”

AA: Proper citations were added.

RV: 2. Several abbreviations should be defined before use; ECG, EMG, TP, TN, FN, FP, MLP

AA: We added definitions of abbreviations mentioned above. We made sure that definitions were provided before the first appearance of abbreviation.

RV: 3. Certain sentences require commas to read better. Grammatical errors should be corrected; line 27-28, 58-60, 67-68, 73-75,90-91,138-139, 311-312, 469 (This is not an exhaustive list. Authors are advised to re-read the manuscript)

AA: All lines pointed out were checked and corrected. We have re-read the manuscript, corrected errors, and also asked one of our colleague for a fresh look.

RV: 4. Section 1.1 and section 1.2 are both discussed as feature extraction methods. Section 1.1 should therefore be renamed appropriately perhaps as Independent Component Analysis

AA: Introduction structure was changed. We believe that the current names of subsections in the introduction should be more informative.

RV: 5. Consistency should be maintained. Equation (17) D is undefined. Does it refer to the width of the window which was defined as d in the text? Ensure all variables used in equations are defined e.g Df in eqn 24 & 25.

AA: D is the width of the window. Naming was changed to improve consistency. All equations were checked for missing references.

RV: 6. Justification for using negentropy should be included.

AA: Justification for negentropy usage was added to text.

RV: 7. Line 276 ci?

AA: This is some error introduced during editing of our article. It was removed.

RV: 8. Table 5 and 6 should be swapped to correspond with the order in which they are discussed in the text to reduce confusion.

AA: I am sorry, but there was no Table 6 in the submitted paper. There is one reference to table 6, but this should be in fact reference to figure 6 (previously data in these plots were presented as a table). This error was corrected. Sorry for confusion. All other references were checked and corrected if needed.

RV: 9. Table headings could be more descriptive e.g “Table 5 Best MLP network for heart beat classification compared to selected convolutional networks reported in the literature”

AA: Thank you very much for your attention. We have supplemented the descriptions for the tables. 

RV: 10. Authors argue that in low data regimes the proposed method is comparable to convolutional neural networks, this could be included in the comparison tables for ease of comparison.

AA: Information about performance for classes with low number of learning examples is presented in Figure 9. For clarity we included information about the number of beats available with a short conclusion. This should justify our claims about performance in low data regimes. 

RV: 11. Justification for some choice of algorithms should be included e.g why stratified 5-fold cross validation, Pan-Tompkins algorithm etc… were these selected because they are commonly used in the literature? Could the technique benefit from other choices?

AA: Pan-Tompkins is commonly used algorithm for finding R wave in ECG signal. Proper mention was added.

A 5-fold cross-validation was used because it ensures the statistical representability of larger data sets. This validation allows for the lowest bias and variance.

Reviewer #2 

Thank you very much for your substantive comments allowing us to improve our article. 

RV: Reviewer 

AA: Author Answer

RV: C1. The abstract shall be rewritten to reflect all results yielded.

AA: Abstract was rewritten to include most important result and conclusions.

RV: C2. The Introduction could be shorter with more concentration on state of the art and the advantage of the proposed methodology.

AA: Introduction was reorganized, and unnecessary information was removed to improve clarity. Subsection with ECG classification methods was rewritten to contain more recent approaches. 

RV: C3: In any case, profound grammatical revisions are needed to render the document useful for readers.

AA: Document was checked for grammar, readability and clarity. We belive that major errors were removed and article in current form should be easier to follow.

RV: C4: The article should have clear arguments regarding the proposed method's rationale and motivations and explicit discussions of the advances versus the prior art.

AA: Separate subsection for aims and contributions in Introduction was created to highlight goals of our work and possible benefits of our method. Detailed discussion of benefits and obtained results is given in section 4.

---

## [Decision Letter · Decision Letter 1]

17 Nov 2021

Novel Feature Extraction Method for Signal Analysis based on Independent Component Analysis and Wavelet Transform

PONE-D-21-24171R1

Dear Dr. Topolski,

We’re pleased to inform you that your manuscript has been judged scientifically suitable for publication and will be formally accepted for publication once it meets all outstanding technical requirements.

Kind regards,

Mahmoud Al Ahmad, PhD

Academic Editor

PLOS ONE

Additional Editor Comments (optional):

Reviewers' comments:

Reviewer's Responses to Questions

**Comments to the Author**

1. If the authors have adequately addressed your comments raised in a previous round of review and you feel that this manuscript is now acceptable for publication, you may indicate that here to bypass the “Comments to the Author” section, enter your conflict of interest statement in the “Confidential to Editor” section, and submit your "Accept" recommendation.

Reviewer #1: All comments have been addressed

Reviewer #2: All comments have been addressed

2. Is the manuscript technically sound, and do the data support the conclusions?

Reviewer #1: (No Response)

Reviewer #2: Yes

3. Has the statistical analysis been performed appropriately and rigorously? 

Reviewer #1: (No Response)

Reviewer #2: Yes

4. Have the authors made all data underlying the findings in their manuscript fully available?

Reviewer #1: (No Response)

Reviewer #2: Yes

5. Is the manuscript presented in an intelligible fashion and written in standard English?

Reviewer #1: (No Response)

Reviewer #2: Yes

6. Review Comments to the Author

Reviewer #1: (No Response)

Reviewer #2: After considering all of the raised comments , I recommend to accept this article to be published in PLOS ONE journal.

7. PLOS authors have the option to publish the peer review history of their article (what does this mean?). If published, this will include your full peer review and any attached files.

Reviewer #1: No

Reviewer #2: No

---

## [Editor Report · Acceptance letter]

23 Nov 2021

PONE-D-21-24171R1 

Novel Feature Extraction Method for Signal Analysis based on Independent Component Analysis and Wavelet Transform 

Dear Dr. Topolski:

I'm pleased to inform you that your manuscript has been deemed suitable for publication in PLOS ONE. Congratulations! Your manuscript is now with our production department. 

Kind regards, 

on behalf of

Dr. Mahmoud Al Ahmad 

Academic Editor

PLOS ONE